# Analysis of Bacteriohopanoids from Thermophilic Bacteria by Liquid Chromatography–Mass Spectrometry

**DOI:** 10.3390/microorganisms9102062

**Published:** 2021-09-30

**Authors:** Irena Kolouchová, Elizaveta Timkina, Olga Maťátková, Lucie Kyselová, Tomáš Řezanka

**Affiliations:** 1Department of Biotechnology, Faculty of Food and Biochemical Technology, University of Chemistry and Technology Prague, Technická 5, 16 628 Prague, Czech Republic; irena.kolouchova@vscht.cz (I.K.); Elizaveta.Timkina@vscht.cz (E.T.); olga.matatkova@vscht.cz (O.M.); 2Research Institute of Brewing and Malting, Lípová 511, 120 44 Prague, Czech Republic; kyselova@beerresearch.cz; 3Institute of Microbiology, The Czech Academy of Sciences, Vídeňská 1083, 142 20 Prague, Czech Republic

**Keywords:** hopanoids, thermophilic bacteria, *Geobacillus stearothermophilus*, liquid chromatography–high-resolution electrospray mass spectrometry

## Abstract

**Background**: Hopanoids modify plasma membrane properties in bacteria and are often compared to sterols that modulate membrane fluidity in eukaryotes. In some microorganisms, they can also allow adaptations to extreme environments. **Methods**: Hopanoids were identified by liquid chromatography–mass spectrometry in fourteen strains of thermophilic bacteria belonging to five genera, i.e., *Alicyclobacillus, Brevibacillus, Geobacillus, Meiothermus*, and *Thermus*. The bacteria were cultivated at temperatures from 42 to 70 °C. **Results**: Regardless of the source of origin, the strains have the same tendency to adapt the hopanoid content depending on the cultivation temperature. In the case of aminopentol, its content increases; aminotetrol does not show a significant change; and in the case of aminotriol the content decreases by almost a third. The content of bacteriohopanetetrol and bacteriohopanetetrol glycoside decreases with increasing temperature, while in the case of adenosylhopane the opposite trend was found. **Conclusions**: Changes in hopanoid content can be explained by increased biosynthesis, where adenosylhopane is the first intermediate in the biosynthesis of the hopanoid side chain.

## 1. Introduction

The geographic unit known as Eger Graben, after the river Eger, is located in the NW part of the Bohemian Massif (Czech Republic). It is elongated in the SW–NE direction along the Ore Mts (Erzgebirge) mountain chain and forms a part of the European Cenozoic rift system. The origin of this large, asymmetric, tectonic graben was related to the Alpine orogeny, which was manifested by tectonic and volcanic activity in the orogen’s foreland [1]. The volcanism began in the Late Cretaceous and reached its peak in the Tertiary period. Owing to this volcanic activity, Karlovy Vary (Carlsbad) has become a world-famous spa. There are several mineral springs in the town of Carlsbad, the main being “Vřídlo”—the official name for a spring with a yield of 2000 L per minute and a temperature of 73 °C. The presence of thermophilic bacteria, such as bacteria of the genera *Thermus* [2] or *Brevibacillus* and *Geobacillus* [3], was discovered.

Several different definitions have been used for thermophilic and thermotolerant microorganisms. The most commonly used one depends on the optimal growth temperature, i.e., an interval of 45 to 80 °C for thermophiles [4].

Prokaryotic microorganisms, such as bacteria, but also cyanobacteria, with a few exceptions, do not contain sterols commonly found in plants and animals. However, many of them contain bacteriohopanepolyols, which are complex molecules based on the hopane skeleton (pentacyclic triterpenoid). The analysis of bacteriohopanoids directly from bacteria is not a well-studied topic. So far, they have been identified in only a few cases (see Appendix A), mainly in *Proteobacteria*. Published examples include, e.g., *Alicyclobacillus acidocaldarius* [5], wild-type *Streptomyces coelicolor* (*Actinobacteria*), which produces aminobacteriohopanetriol [6], or in the paper of Hippchen et al. [7], where two bacteriohopanoids (l-(*O*-β-*N*-acylglucosaminyl)-2,3,4-tetrahydroxypentane-29-hopane and 1,2,3,4-tetrahydroxypentane-29-hopane) were identified from *Bacillus acidocaldarius* (now *Alicyclobacillus acidocaldarius*), which belongs to the *Firmicutes* division.

With the growing number of microorganisms for which the genome is known, a key enzyme for the synthesis of hopanoids has been identified as squalene-hopene cyclase (EC 5.4.99.17). Squalene is the primary precursor of hopane skeleton biosynthesis. Squalene cyclization proceeds without the migration of methyl groups to form C30 diploptene or diplopterol and does not depend on the presence of oxygen. Therefore, bacterial hopanoids lack the 3β-hydroxyl group of sterols. The synthesis occurs in a single joint reaction by squalene-hopene cyclase. Possible modifications of the skeleton of hopanoids can be divided into the addition of a methyl group to form 2β- and/or 3β-methyl hopanoids and further into the modification (construction) of the side chain. Adenosylhopane is the first intermediate produced during the biosynthesis of bacteriohopanepolyol side chains.

Furthermore, adenosylhopane loses its adenyl group to form ribosylhopane, and the D-ribose moiety can open to produce bacteriohopanetetrol with four hydroxyls in the side chain. Further modifications can be made to the terminal hydroxyl group of the side chain, which may be variously substituted, for example, by a glycoside or ether linkage to glucosamine or ribose. In some species, the polyhydroxyl side chain may be acylated. Other hydroxylated forms are known, e.g., penta- or hexaols and amino polyols occur in some genera.

The use of hopanoids or their 2- and/or 3-methyl derivatives as biological markers for specific microorganisms in geological formations from recent sediments to oil deposits was proven to not be fully functional. Instead, they may be indicators of origin in a particular environment, such as microbial communities or microbial mats in hot springs [8].

Hopanoids are thought to have very similar functions to sterols in eukaryotic membranes. They can intercalate into phospholipid bilayers and modulate membrane fluidity by interacting with their complex lipid components. Hopanoids are important in reducing membrane permeability, including oxygen diffusion, and adapting to stress caused by extreme environmental conditions, including high temperatures. *Alicyclobacillus acidocaldarius* has been reported to synthesize several times more hopanoids at 65 °C than at 60 °C, probably due to increased lipid membrane fluidity [9]. In a model membrane composed of bacteriohopanetetrol ((32R,33S,34S)-bacteriohopane-32,33,34,35-tetrol) and phosphatidylcholine, the resulting effect is strongly temperature-dependent [10].

In the analysis of complex mixtures of bacteriohopanoids, the polar groups (-OH and/or -NH_2_) are usually derivatized (acetylated) and subsequently identified by either GC/MS [11] or, even better, LC–MS [12,13,14,15,16,17,18]. A paper describing the analysis of non-derivatized bacteriohopanepolyols by LC–MS (ultrahigh-performance liquid chromatography/positive ion atmospheric pressure chemical ionization (UPLC/APCI)) has been published [19]. Nevertheless, based on data from the literature [20] and our previously published paper [21], we used the LC–MS method for determination of derivatized bacteriohopanoids. Thermophilic and thermotolerant strains of bacteria were cultured at various temperatures, from 42 to 70 °C, and analysis was performed by LC–MS and tandem MS, respectively, using a precursor ion (PIS) and neutral ion scans (NLS).

## 2. Materials and Methods

### 2.1. Chemicals and Standards, Isolation, Molecular Identification, and Cultivation of Four Strains of Thermophilic Bacteria from Hot Springs

All chemicals were purchased from Merck (Darmstadt, Germany).

Four strains of thermophilic bacteria were isolated from thermal springs Štěpánka, Sadový, Mlýnský, and Vřídlo in Carlsbad, Czech Republic, and are listed in Appendix A (see Appendix A and a previously published paper [3]). Water samples were taken from thermal springs using 50 mL sterile Falcon tubes in three replicates for each sample from the same spring of the same location, and were transported to the laboratory at a temperature of at least 50 °C, ensured by a thermal box. The collected water samples were immediately used for enrichment cultivations as follows: volume of 15 mL of spring water was supplemented with 1.5 mL of diluted YPD (dextrose 20 g/L, peptone 20 g/L, and yeast extract 10 g/L, pH 7.0) and cultivated on a rotary shaker (150 rpm). Cultivations were performed at the temperature corresponding to the original spring source (i.e., 58 °C in Štěpánka, Mlýnský, and Vřídlo, and 42 °C in Sadový). After 30 days of cultivation, aliquots of the enrichment cultures were streaked on an agar medium to obtain separate colonies. Reasoner’s 2A agar (R2A, HiMedia, Brno, Czech Republic) and Thermus 162 agar (HiMedia, Brno, Czech Republic) supplemented by micro-filtered origin spring water were used. Colonies with different morphologies were further cultivated and identified by 16S rRNA as described before [3]. Four bacterial isolates, designated *Geobacillus stearothermophilus* ST-YPD, *Brevibacillus agri* SA-1, *Geobacillus kaustophilus* ML-1, and *Geobacillus stearothermophilus* VR-1, were obtained from the springs. For pre-inoculum preparation, the strains were cultivated in Luria–Bertani broth (LB: tryptone 10 g/L, yeast extract 5 g/L, and NaCl 10 g/L, pH 7.0) on an orbital shaker (150 RPM) for 72 h. Cultivation of biomass for further analysis was carried out in LB broth. Two hundred milliliters of sterile LB broth was inoculated with pre-inoculum to achieve OD600 0.2 and incubated on an orbital shaker (150 RPM) for 48 h until the stationary phase was reached. The isolate *Geobacillus stearothermophilus* ST-YPD was cultivated at 42 °C and the remaining three isolates at 58 °C (in conformity with the temperature of the springs of origin). At the end of cultivation, cells were harvested by centrifugation (10,000× *g*, 10 min, 4 °C) and washed twice with sterile 0.9% NaCl. Cell mass was frozen at −70 °C and lyophilized.

### 2.2. Extraction and Isolation of Hopanoids

The extraction procedure was based on the method of Bligh and Dyer [22]. Briefly, the lyophilized cells (approximately 10 mg) were suspended in a chloroform–methanol mixture (2:1) (~1 mL) for 30 min at 20 °C with stirring, after which chloroform and water were added and the insoluble material was separated by centrifugation. The aqueous phase was aspirated off, and the chloroform phase was evaporated to dryness under reduced pressure. Lipid extracts were acetylated with acetic anhydride in pyridine (0.5 mL, 1:1 *v*/*v*) for one hour at 50 °C and left at room temperature overnight. The solvent was removed under a vacuum. The acetylated extract was dissolved in acetonitrile (ACN):propan-2-ol (iPrOH) (1:1 *v*/*v*) and used for LC–MS analysis.

The lyophilized cells of the previously cultured strains listed in Appendix A were extracted and further analyzed in a similar manner as the four strains mentioned in Appendix A.

### 2.3. Analysis of Hopanoids by LC–MS

The LC equipment consisted of a 1090 Win system, PV5 ternary pump, an automatic injector (HP 1090 series, Agilent, Santa Clara, CA, USA), and three Luna Omega 1.6 µm, reverse-phase (RP) C18, 100 Å, LC Columns L × I.D. 150 × 2.1 mm, connected in series. Although the efficiency stated by the manufacturer was not achieved, i.e., 350,000 theoretical plates/meter (see manufacturer’s literature), the efficiency achieved in our case was only ~115,000 theoretical plates/meter. Pregnane-3,20-diol 3,20-diacetate with a retention time (tR) 23.9 min was used as an external standard, with a flow rate of 0.95 mL/min, an injection volume of 10 μL, and a column temperature of 25 °C. Peracetylated hopanoids were separated using a solvent gradient program with ACN and iPrOH: initial ACN/iPrOH (85:15, vol/vol), a linear change to 15:85 ACN/iPrOH (vol/vol) from 0 min to 60 min, and a hold for 10 min. The composition was returned to the initial conditions over 10 min. For ESI-MS analysis, an LTQ Orbitrap Velos was used.

An LTQ Orbitrap Velos mass spectrometer (Thermo Fisher Scientific, San Jose, CA, USA), a high-resolution hybrid mass spectrometer equipped with a heated electrospray interface (H-ESI), was operated in the positive ionization mode. The MS scan range was performed in the Fourier transform mode and recorded within a window between 150–1500 *m*/*z*. Mass spectra were acquired with a target mass resolution of R = 110,000 at *m*/*z* 800, and the ion spray voltage was set at 3.5 kV in the positive ionization mode. For positive ionization mode the following parameters were used: sheath gas flow, 18 arbitrary units (AU); auxiliary gas flow, 7 AU; ion source temperature, 250 °C; capillary temperature, 230 °C; capillary voltage, 50 V; and tube lens voltage, 170 V. Helium was used as a collision gas for collision-induced dissociation (CID) experiments. The CID normalization energy of 35% was used for the fragmentation of parental ions. The MS/MS product ions were detected in high-resolution FT mode. The calibration of the MS spectrometer was conducted using a Pierce LTQ Orbitrap Positive Ion Calibration Solution (Thermo Fisher Scientific, San Jose, CA, USA). The mass accuracy was better than 0.9 ppm. The chemical structures of compounds were confirmed with the help of the spectral database LIPID MAPS^®^ Lipidomics Gateway (http://www.lipidmaps.org/, (accessed on 14 September 2021)).

### 2.4. Quantification

Pregnane-3,20-diol-3,20-diacetate was obtained from commercially available pregnanediol and was acetylated at room temperature overnight with pyridine/acetic anhydride (1:1; 2 mL). A semi-quantitative estimate of hopanoid abundance was calculated from the characteristic base peak ([M + H-CH_3_COOH]^+^) of individual hopanoids relative to the *m*/*z* 345.2788 (base peak) of the acetylated 5α-pregnane-3β,20β-diol internal standard. The limit of detection (LOD) was ~40 pg (0.1 pM) at S/N = 3 and the limit of quantification (LOQ) at S/N = 10 was about 132 pg. The content of additional bacteriohopanepolyols was then calculated from the relative amount in the HPLC analysis. Without standards for each hopanoid, a significant potential error in quantification requires caution when comparing these values with those in the literature.

### 2.5. Statistical Analysis

The statistical analysis was performed using the IBM SPSS Statistics (Statistical Package for the Social Sciences; IBM Corp, 2013) software, version 26 (IBM^®^ Corporation, Armonk, NY, USA).

## 3. Results and Discussion

### 3.1. Characterization and Identification of Bacterial Isolates

Molecular classification of the four strains collected from hot springs of Carlsbad was performed by 16S rRNA gene sequence analysis, as shown in Appendix A. The results of the molecular analysis were consistent with the phenotypic characteristics [3]; therefore, the strains were designated *Brevibacillus agri* SA-1, *Geobacillus kaustophilus* ML-1, *Geobacillus stearothermophilus* ST-YPD, and *Geobacillus stearothermophilus* VR-1.

### 3.2. Analysis of Total Hopanoids by Liquid Chromatography–Mass Spectrometry Using Precursor Ion and Neutral Ion Scans

According to Bligh and Dyer, total lipids obtained from fourteen bacteria cultured at different temperatures [22] were acetylated and analyzed by RP-UPLC/high-resolution positive tandem ESI. Hopanoids were derivatized (acetylated) mainly for reasons of reducing the polarity of individual compounds on the basis of already-published papers [13,14,15,16,17,18]. In essence, there is only one paper [19] which describes the analysis of non-derivatized hopanoids. Conversely, many more papers (see publications from Talbot lab) describe the analysis of derivatized hopanoids. Bacteriohopanoid standards are not commercially available, and they have therefore not been tested. Based on our previous experience [23], we used a completely different elution mixture than that described in various papers [15,16,19]. This was mainly due to the fact that the elution mixture with acetonitrile has a lower viscosity than the commonly used methanol–water–isopropanol mixture. Lower viscosity of the elution mixture used by us made it possible to use a combination of three columns in a series and thus achieve an efficiency of ~115,000 theoretical plates. This value is at least an order of magnitude higher than the column efficiency, for example, for *N*-palmitoleyl of peracetylated bacteriohopanetriol (Figure 4 in paper [15]). Due to this resolution, it was not a problem to separate individual molecular species and isomers, see for example the compounds with tR 35.06 vs. 35.45, 38.11 vs. 38.63, and 42.67 vs. 43.03 min, respectively (Appendix A). The presence of stereoisomers of hopanoids is quite common in bacteria [18,21,24]. The high-resolution NLS and PIS of the peracetylated total lipid extract made it possible to identify twenty-one molecular species (see Figure 1). Furthermore, the use of tandem MS confirmed the structure of *N*-acyl-substituted bacteriohopanepolyols, as shown in Appendix A.

### 3.3. The Absence of Methylhopanoids in the Studied Bacteria was Proven by Liquid Chromatography–Mass Spectrometry

Figure 1 shows the PIS at *m*/*z* 205.1951 and NLS at *m*/*z* 206.2035. If methyl-substituted hopanoids were to be present, then both PIS and NLS would have to show peaks. As shown in Figure 1, no peaks are present; hence, their concentration is at least three orders of magnitude lower than the concentration of non-methyl hopanoids. The presence of genes encoding the production of enzymes, i.e., the two methylases required to produce 2- or 3-methylhopanoids, has been shown in the literature [25]. The presence of both hopanoid C-2 methylase (EC: 2.1.1.-) and/or hopanoid C-3 methylase (EC: 2.1.1.-) in the studied bacterial species was analyzed using the BLASP tool from the NCBI. All species, except for *Meiothermus ruber* and *Thermus aquaticus*, possess a protein sequence of high similarity to the enzymes mentioned above; see Appendix A for more information. Nevertheless, the presence of a gene does not require its expression or metabolite biosynthesis.

The tandem mass spectra of both methyl derivatives (2β-methyl and 3β-methyl hopanoids, respectively) are characterized by a fragment ion *m*/*z* 205.1951 instead of 191.1794, and a series of fragments resulting from a characteristic neutral loss of 206.2035 Da instead of 192.1878 Da (as a result of C-9 cleavage in C and 11 and C-8 and 14) [17]. These *m*/*z* values are ~14 units higher than those of non-methyl bacteriohopanepolyols.

The compounds of the 2β-methylhopanoid type are present in cyanobacteria [26,27,28] although Rashby et al. [29] found 2-methylbacteriohopanetetrol in the anoxygenic phototroph *Rhodopseudomonas palustris*. Methylated C-3 hopanoids have been detected in a limited number of bacteria. 3β-methylated tetra-, penta- and hexafunctionalized hopanoids have been identified in many lake sediments, including the world-famous Loch Ness. Methanotrophs and acetic acid bacteria produce these 3β-methylhopanoids [26,30]. 3-methylhopanoids have historically been associated with aerobic methanotrophy, based on culture experiments [31] and co-occurrence with aerobic methanotrophs in the environment [32]. They are also present in some *Acidobacteria* [33].

### 3.4. Use of Two Tandem Mass Spectrometry Methods (PIS and/or NLS) for the Identification of Hopanoid Structures

The major ions in the tandem MS spectra of compounds (**1b**–**e**, **2e**, and **3e**) come from the partial or complete loss of up to five acetylated hydroxyl groups (Figure 2 and Table 1). An ion of the structure, [M + H-CH_3_COOH]^+^, was identified as the base peak in all tandem MS. The key ion confirming the amide structure was the ion of the type [M + H-CH_3_COOH-RC=O]^+^; depending on the number of OH groups, values at 728.5100, 668.4888, and 608.4677 Da were found. It was also shown by tandem MS that ring A is not substituted with a methyl group (see above).

An example is the tandem MS of the compound triacetate (**4**) at *m*/*z* 788.5323 [M + H]^+^. The tandem MS spectrum (see Table 2) dominates the base peak at *m*/*z* 611.4678, produced by adenine loss. Less important ions at *m*/*z* 551.4466 and 491.4249 result from the loss of acetylated hydroxyl groups, and ion at *m*/*z* 419.2799 indicate a hopanoid ring system resulting from fission in ring C and losses of 192.1880 Da from ions at *m*/*z* 611.4678.

Bacteriohopanetetrol (**5**) is usually present in all samples obtained from both bacteria and sediments. The protonated molecule of the fully acetylated bacteriohopanetetrol ([M + H]^+^ at *m*/*z* 715.5145) caused a rapid loss of one acetate and formed an ion at *m*/*z* 655.4940 ([M + H-CH_3_COOH]^+^), which is the base peak in tandem MS. Based on this value, it can be judged that nitrogen is not present in the molecule but is replaced by oxygen. The tandem MS also contains other ions indicating the loss of functional groups, i.e., CH_3_COOH (60.0211 Da) and ions showing the loss of the A + B rings, which was confirmed by an ion at *m*/*z* 191.1794.

The basic peak in the tandem mass spectrum of bacteriohopanetetrol glycoside (formula **6**) is the ion with the structure [M + H-CH_3_COOH]^+^. The ion at *m*/*z* 1002.6151 is a protonated molecule [M + H]^+^ due to the presence of an amino group at the terminal group. Three ions are present in the tandem MS, resulting in a gradual loss of acetate at *m*/*z* 942.5944, 882.5733, and 822.5522. The ion at *m*/*z* 655.4940 results from the loss of the terminal group due to a cleavage between C-35 and the oxygen at C-35, indicating the tetrafunctionalized side chain. The ion at *m*/*z* 330.1191 corresponds to the terminal group following the cleavage between oxygen at C-35 and C-1’ on the terminal group.

Two *N*-acylaminotriols containing palmitic and palmitoleic fatty acids bound to aminotriol via an amide bond have been previously identified in *Nitrosomonas europaea* [15,34]. A total of five *N*-acyl derivatives were identified, i.e., two saturated (palmitic and stearic acids) and three unsaturated (hexadecenoic, octadecenoic, and nonadecenoic acids). Furthermore, unsaturated C18 *N*-acylaminotriol was preliminarily identified in the purple non-sulfur bacterium *Rhodomicrobium vannielii* [35].

### 3.5. Influence of Cultivation Temperature on the Structure and Abundance of Hopanoid´s Molecular Species

Table 3 shows the content of all identified hopanoids depending on the culture temperature. An example is a strain of *G. stearothermophilus*, which has been cultured at three different temperatures (55, 58, and 70 °C). Regardless of the source of origin (two different hot springs or evaporated milk), all five strains show the same tendency to change the content of hopanoids. In the case of aminopentol (**1a**) its content increases, aminotetrol (**2a**) shows no substantial change, and in the case of aminotriol (**3a**), the content decreases by almost a third. Similar results were obtained with *M. ruber*, which was cultured at 55, 65, and 70 °C. Again, the content of aminopentol increased, did not change with aminotetrol and decreased with aminotriol. The remaining two strains, i.e., *A. acidoterrestris* and *T. aquaticus*, show essentially the same trend. The situation with *N*-acyl derivatives is far more complicated. In this case, the *N*-acyl chain changes the molecule’s shape (see Appendix A), and thus, of course, the spatial arrangement of lipids in the membrane.

Especially in all cultured thermophilic bacteria, the proportion of even and odd *N*-acyls is the same. In many cases, the molecular species having *N*-acyls with odd numbers of carbon atoms are more abundant. Furthermore, there is also a general trend that the proportion of *N*-acyl pentols increases, and the proportion of *N*-acyl triols decreases with increasing temperature. The three hopanoids, i.e., molecular species with structures **4**, **5,** and **6**, differ from each other depending on the content and temperature. While the content of compounds **5** and **6** decreases with increasing temperature, the opposite is true for adenosylhopane (**4**). This large abundance can be explained, for example, by increased biosynthesis, where adenosylhopane is the first intermediate in hopanoid side chain biosynthesis [36].

Hot terrestrial springs provide suitable habitats for several photoautotrophic and chemolithoautotrophic microorganisms. The distribution of these organisms seems to be mostly affected by temperature. For example, temperatures above 73 °C often eliminate the growth of cyanobacteria and other phototrophic bacteria and, conversely, allow the growth of chemolithoautotrophs. The effect of environmental conditions on the production of bacteriohopanepolyols has been described in several papers (see below). These changes occur because hopanoids regulate membrane fluidity and induce the order in the phospholipid membrane [10,37,38], which helps to protect the microorganism from external stresses.

A study was published on symbiotic methanotrophs in *Sphagnum* moss [39,40], where the proportion of aminotriol decreased with increasing temperature (from 5 to 25 °C), while the abundance of both aminotetrol and aminopentol grew. Similarly, Osborne et al. [41] describe changes in the abundance of bacteriohopanepolyols as a function of temperature. It was shown that with increasing temperature (from 4 °C to 40 °C) aminotriol also decreased, while the presence of aminopentol increased and the aminotetrol content remained constant. Jahnke et al. [42] report that for the psychrotolerant methanotroph (considered *Methylomonas methanica*), the relative percentage of C31 hopanol decreased. In contrast, C30 hopanol increased with increasing temperature. Similarly, derivatives of hopanols after the oxidation of hopanoids have been analyzed in other methanotrophic bacteria of the genus *Methylovulum* [43]. Again, it was confirmed that the content of pentols increased, the triol content decreased, and the tetrol content remained more or less constant as the cultivation temperature increased. Unfortunately, all the publications mentioned above analyzed the degradation products formed by the oxidation of vicinal diols (Rohmer degradation) and, of course, quantified. Although this procedure dramatically simplifies the analysis of hopanoids, the information about the structure of native hopanoids is completely lost.

Previous studies show that the proportion of total hopanoids in the thermoacidophilic bacterium *Alicyclobacillus acidocaldarius* increases with culture temperature. As the temperature changed from 60 to 65 °C, the content of tetrol (1,2,3,4-tetrahydroxypentane-29-hopane) increased several times regardless of the pH of the culture. Studies [9,44] have shown that a change in culture temperature from 14 °C to 37 °C in ethanologenic *Zymomonas mobilis* was accompanied by an increase in 1,2,3,4-tetrahydroxypentane-29-hopane by several tens of percent. Additionally, Joyeux et al. [45] investigated the effect of culture temperature on hopanoid content in acetic acid bacterium *Frateuria aurantia*. It was found that the C31/C32 ratio of hopanoid degradation products was more than doubled in the culture performed at 36 °C compared to the culture at 16 °C.

Mono- and di-unsaturated bacteriohopanols were not identified in the thermophilic bacteria analyzed in this paper. One of the possible hypotheses of their absence is that they contribute to the increase in membrane fluidity, which is undesirable in thermophilic bacteria. These compounds were isolated from *Gluconacetobacter xylinus*, *Burkholderia cepacia*, methane-oxidizing *Methylocaldum szegediense* [46], or *Rhodopseudomas palustris* [47].

## 4. Conclusions

Although the enzyme squalene-hopene cyclase is cited more than 26,000 times in a GenBank search, only an estimated 10% of bacteria can produce hopanoids [48,49]. Even if many phyla of bacteria are potentially capable of producing hopanoids, their production was detected by either GC–MS or LC–MS in only a few dozen (see Appendix A of individual strains that were cultured).

Twenty-one intact bacteriohopanepolyols in extracts from fourteen strains of bacteria were identified by RP-UPLC/HR-ESI^+^-MS/MS. The distribution varied depending on the strain of bacteria and also the cultivation temperature. Regardless of the place of origin, all five strains of *Geobacillus stearothermophilus* have the same tendency to change the hopanoid content. In the case of aminopentol (**1a**) its content increases, aminotetrol (**2a**) does not show a significant change, and in the case of aminotriol (**3a**) the content decreases by almost a third. Similar results were obtained with *Meiothermus ruber*, which was cultured at 55, 65, and 70 °C. In the case of *N*-acyl derivatives, the proportion of *N*-acylpentols increases and the proportion of *N*-acyltriols decreases with increasing temperature. The content of bacteriohopanetetrol and bacteriohopanetetrol glycoside decreases with increasing temperature, while in the case of adenosylhopane it is exactly the opposite. This dependence can be explained, for example, by increased biosynthesis, where adenosylhopane is the first intermediate in the biosynthesis of the hopanoid side chain.

This study expands the knowledge of the high diversity of hopanoid-producing bacteria in various hot springs, and suggests that bacteriohopanepolyols can be detected in individual cultures. The analysis of bacteriohopanepolyols by LC–MS, especially by RP-UPLC/HR-ESI^+^-MS/MS, is a valuable tool for evaluating hopanoids in bacteria living in hot springs.

## Figures and Tables

**Figure 1 microorganisms-09-02062-f001:**
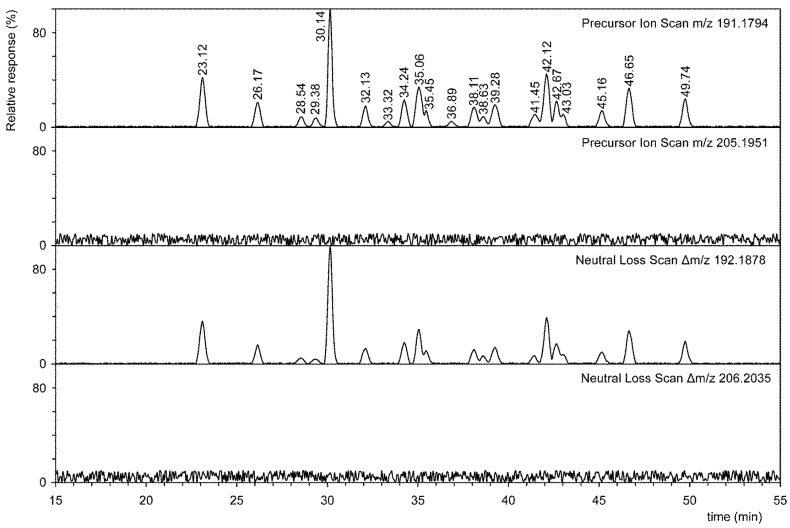
Liquid chromatography–mass spectrometry of the peracetylated hopanoids using high resolution NLS and PIS.

**Figure 2 microorganisms-09-02062-f002:**
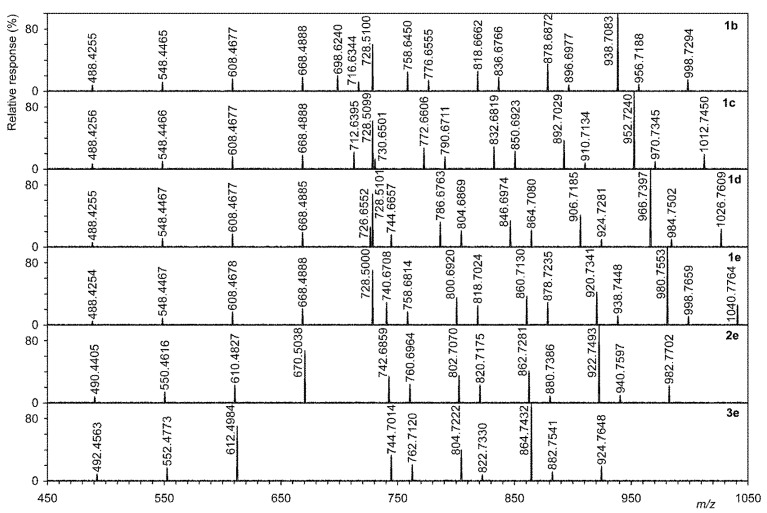
The major ions in the tandem MS spectra of compounds (**1b–e**, **2e**, and **3e**).

**Table 1 microorganisms-09-02062-t001:** Assignment of ions observed in tandem high-resolution electrospray mass spectra of hopanoids described in the text.

	1b		1c		1d		1e		2e		3e	
Ion Description	*m*/*z*	Abund ^a^	*m*/*z*	Abund	*m*/*z*	Abund	*m*/*z*	Abund	*m*/*z*	Abund	*m*/*z*	Abund
[M + H]^+^	998.7294	15	1012.7450	19	1026.7609	23	1040.7764	26	982.7702	22	924.7648	19
[M + H-COCH_2_]^+^	956.7188	9	970.7345	10	984.7502	10	998.7659	11	940.7597	10	882.7541	12
[M + H-CH_3_COOH]^+^	938.7083	100	952.7240	100	966.7397	100	980.7553	100	922.7493	100	864.7432	100
[M + H-CH_3_COOH-COCH_2_]^+^	896.6977	8	910.7134	8	924.7281	10	938.7448	12	880.7386	9	822.7330	8
[M + H-2 × CH_3_COOH]^+^	878.6872	35	892.7029	37	906.7185	41	920.7341	42	862.7281	41	804.7222	40
[M + H-2 × CH_3_COOH-COCH_2_]^+^	836.6766	18	850.6923	23	864.7080	22	878.7235	29	820.7175	23	762.7120	21
[M + H-3 × CH_3_COOH]^+^	818.6662	26	832.6819	29	846.6974	34	860.7130	37	802.7070	35	744.7014	33
[M + H-3 × CH_3_COOH-COCH_2_]^+^	776.6555	14	790.6711	16	804.6869	21	818.7024	25	760.6964	24	-	0
[M + H-4 × CH_3_COOH]^+^	758.6450	25	772.6606	27	786.6763	32	800.6920	35	742.6859	34	-	0
[M + H-4 × CH_3_COOH-COCH_2_]^+^	716.6344	12	730.6501	13	744.6657	16	758.6814	17	-	0	-	0
[M + H-5 × CH_3_COOH]^+^	698.6240	20	712.6395	22	726.6552	26	740.6708	29	-	0	-	0
[M + H-CH_3_COOH-RC=O]^+^	728.5100	60	728.5099	62	728.5101	68	728.5000	70	670.5038	67	612.4984	70
[M + H-2 × CH_3_COOH-RC=O]^+^	668.4888	18	668.4888	18	668.4885	19	668.4888	21	610.4827	23	552.4773	17
[M + H-3 × CH_3_COOH-RC=O]^+^	608.4677	16	608.4677	16	608.4677	17	608.4678	17	550.4616	14	492.4563	9
[M + H-4 × CH_3_COOH-RC=O]^+^	548.4465	12	548.4466	10	548.4467	11	548.4467	9	490.4405	8	-	0
[M + H-5 × CH_3_COOH-RC=O]^+^	488.4255	8	488.4256	7	488.4255	6	488.4254	5	-	0	-	0
[A+B cycles]^+^	191.1796	13	191.1795	15	191.1796	16	191.1794	18	191.1796	11	191.1795	9

^a^ Relative %, relative to base peak.

**Table 2 microorganisms-09-02062-t002:** Assignment of ions observed in tandem mass spectra of peracetylated adenosylhopane described in the text.

Structure of Ion	*m*/*z*
[M + H]^+^	788.5323
[M + H-terminal group + H]^+^	611.4678
[M + H-terminal group-CH_3_COOH]^+^	551.4466
[M + H-terminal group-CH_3_COOH-COCH_2_]^+^	509.4355
[M + H-terminal group-2 × CH_3_COOH]^+^	491.4249
[M + H-terminal group-A + B rings, i.e., ring C cleavage]^+^	419.2799
[M + H-terminal group-A + B rings-CH_3_COOH]^+^	359.2583
[M + H-terminal group-A + B rings-2 × CH_3_COOH]^+^	299.2377

**Table 3 microorganisms-09-02062-t003:** Content (relative %) of hopanoids from fourteen strains of thermophilic bacteria identified by reverse-phase liquid chromatography–positive electrospray mass spectrometry (see Materials and Methods).

Cultiv. Temp	Bacterium	1a	1b	1c	1d	1e	2a	2b	2c	2d	2e	3a	3b	3c	3d	3e	4	5	6
55	*G. stearothermophilus* CCM 2062	5.5	1.3	4.5	2.3	2.9	5.2	0.3	2.9	1.6	1.9	14.7	2.6	10.7	9.1	6.8	12.8	14.0	1.0
58	***G. stearothermophilus* ST-YPD** ^a^	7.7	1.9	5.2	3.5	4.1	5.2	1.1	3.9	1.9	2.5	9.6	1.6	9.1	5.8	5.7	6.3	23.5	1.4
58	***G. stearothermophilus* VR-1** ^a^	7.4	1.7	5.1	3.7	3.9	5.6	0.8	3.9	2.2	3.1	9.3	1.7	9.2	5.6	5.5	8.1	21.8	1.4
70	*G. stearothermophilus* CCM 5965	9.1	2.0	7.3	4.4	4.7	4.9	1.3	4.2	2.7	2.9	5.3	1.2	6.9	4.2	5.5	4.2	27.4	1.8
70	*G. stearothermophilus* CCM 2062	9.3	2.0	7.6	4.2	4.9	4.7	1.1	3.8	2.4	3.2	5.1	1.1	6.6	4.1	5.3	3.9	28.7	2.0
55	*M. ruber* CCM 4212	9.0	1.9	4.6	1.7	6.0	9.7	1.8	2.6	3.1	4.2	6.7	1.6	8.7	5.0	7.9	4.9	19.9	0.8
65	*M. ruber* CCM 4212	11.9	2.1	5.2	2.4	6.3	9.5	1.9	2.6	2.6	3.3	5.5	1.5	7.4	4.9	7.2	4.0	20.6	1.2
70	*M. ruber* CCM 4212	12.0	2.4	6.0	3.0	6.7	9.0	1.7	3.0	3.2	4.1	4.1	1.3	6.2	4.7	6.7	3.2	21.2	1.5
45	*A. acidoterrestris* CCM 4660	8.9	2.6	5.6	4.6	6.6	11.6	2.9	4.0	3.1	5.3	9.6	2.1	3.0	3.4	5.3	4.0	15.2	2.3
70	*A. acidoterrestris* CCM 4660	18.8	3.3	6.3	4.6	7.9	10.1	2.4	3.5	3.0	5.2	3.8	1.3	2.4	3.0	4.9	3.8	13.3	2.4
65	*T. aquaticus* CCM 3488	15.8	3.4	5.1	2.8	5.7	9.4	3.0	3.0	2.3	5.1	10.4	2.1	4.0	2.7	6.4	4.7	10.6	3.4
70	*T. aquaticus* CCM 3488	16.3	3.7	5.8	3.1	6.4	8.8	2.7	3.1	2.0	4.7	8.8	1.7	3.7	2.7	5.8	5.1	11.9	3.7
58	***G. kaustophilus* ML-1** ^a^	11.8	3.0	7.7	2.2	4.4	13.7	3.3	5.9	3.3	6.7	6.3	1.5	5.5	3.7	4.8	3.3	10.7	2.2
42	***B. agri* SA-1** ^a^	10.1	3.6	9.3	2.8	5.6	15.7	2.4	4.4	4.2	6.1	3.6	2.8	3.6	2.8	7.7	2.8	10.9	1.6

^a^ Isolated bacteria in this study are marked in bold (see also Appendix A).

## Data Availability

The acquired sequences (see Appendix A) were submitted to GenBank.

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
