# Peer review of "Analysis of Bacteriohopanoids from Thermophilic Bacteria by Liquid Chromatography–Mass Spectrometry"

_microorganisms, 2021, doi:10.3390/microorganisms9102062_

Round 1
Reviewer 1 Report
The manuscript "Analysis of bacteriohopanoids from thermophilic bacteria by liquid chromatography-mass spectrometry" describes an LC-MS method for the identification of bacteriohopanoids in 14 strains of thermophilic bacteria.
The introduction section is definitely complete, even though it seems a little too lengthy. Moreover, I cannot see properly Figure 1, but I am sure it has to do with the PDF generation rather than the paper itself.
In section 2.1. further details shall be provided.
In results and discussion, an explanation on the need for the derivatization (acetylation) should be added. It is needed for improving the MS ionization or for LC separation? Were the standards tested on the LC-MS system without derivatization?
MS analysis in negative polarity could be helpful in identifying N-acyl derivatives with a higher degree of confidence, due to the likely presence of deprotonated FA ions. Have you tested negative ESI ionization for further structural information?
Author Response
The introduction section is definitely complete, even though it seems a little too lengthy. Moreover, I cannot see properly Figure 1, but I am sure it has to do with the PDF generation rather than the paper itself.
Figure 1 has been removed, see also the request of the second reviewer.
In section 2.1. further details shall be provided.
The following sentences were inserted in the manuscript. “”.
In results and discussion, an explanation on the need for the derivatization (acetylation) should be added. It is needed for improving the MS ionization or for LC separation? Were the standards tested on the LC-MS system without derivatization?
Derivatization (acetylation) is necessary for better separation by LC. Basically, there is only one paper [19] which describes the analysis of non-derivatized hopanoids. Conversely many more papers, see publications from Talbot lab describe the analysis of derivatized hopanoids. Bacteriohopanoids standards are not commercially available and they have therefore not been tested. The manuscript included the sentences “Hopanoids were derivatized (acetylated) mainly for reasons of reducing the polarity of individual compounds on the basis of already published papers [13-18].
MS analysis in negative polarity could be helpful in identifying N-acyl derivatives with a higher degree of confidence, due to the likely presence of deprotonated FA ions. Have you tested negative ESI ionization for further structural information?
We agree with the reviewer that negative ionization could be useful in identifying N-acyl derivatives. It was not performed for two reasons. First, cleavage of the amide bond is not easy. Examples are analogous compounds, i.e. ceramides, where the corresponding ions of the RCOO- type are not formed. This is a great difference from esters, where RCOO- type ions in negative tandem MS have a majority abundance. Second, data from the tandem MS of hopanoids in negative MS mode have not yet been published. This publication is not of an analytical nature and also N-acyl derivatives of bacteriohopanols are not commercially available and therefore we did not use negative mode.
Reviewer 2 Report
The manuscript by Kolouchová et al. describes analysis of bacteriohopanoids from thermophilic bacteria by
liquid chromatography-mass spectrometry. The paper shows interesting insight into influence of temperature on bacteriohopanoids. I only have few comments
- Abstract should be improved (some information and context missing)
- Figure 1 seems to be corrupted
- Titles of chapters in Results and discussion part should be changed to more descriptive
Author Response
Abstract should be improved (some information and context missing)
The abstract has been comprehensively rewritten; see also the request of the second reviewer.
Figure 1 seems to be corrupted
Figure 1 has been removed, see also the request of the second reviewer.
Titles of chapters in Results and discussion part should be changed to more descriptive
The titles of chapters in Results and discussion have been completely rewritten.
Reviewer 3 Report
Kolouchová et al. analyzed bacteriohopanoids from thermophilic bacteria using LC-MS.
The overall quality of the paper is modest.
The Title is representative.
The abstract has to be almost entirely re-written in a more concise and fluent form, abstracts should give a pertinent overview of the work. Follow the standard template in order to offer the reader a clear an easy to comprehend abstract " (1) Background: Place the question addressed in a broad context and highlight the purpose of the study; (2) Methods: briefly describe the main methods or treatments applied; (3) Results: summarize the article's main findings; (4) Conclusions: indicate the main conclusions or interpretations. "
Figure 1 is unacceptable
I can see the authors were very generous with self citations: 3,12, 13,22,23,25,26,27,28,29,30. Again, I find this unacceptable.
2. Materials and Methods
2.1. Chemicals and Standards
" All chemicals were purchased from Merck (Darmstadt, Germany)." This is not relevant as an subchapter
2.2. Isolation, Molecular Identification, and Cultivation of Four Strains of Thermophilic Bacteria from Hot Spring
Line 105: how were the samples collected and transported/stored?
Line 107: How was the isolation of the strains performed?
2.4. Analysis of Hopanoids by LC-MS
Where is the chromatogram? Please add it
Author Response
The abstract has to be almost entirely re-written in a more concise and fluent form, abstracts should give a pertinent overview of the work. Follow the standard template in order to offer the reader a clear an easy to comprehend abstract " (1) Background: Place the question addressed in a broad context and highlight the purpose of the study; (2) Methods: briefly describe the main methods or treatments applied; (3) Results: summarize the article's main findings; (4) Conclusions: indicate the main conclusions or interpretations. "
The abstract has been comprehensively rewritten according to the reviewer's requirements.
Figure 1 is unacceptable
Figure 1 has been removed.
I can see the authors were very generous with self citations: 3,12,13,22,23,25,26,27,28,29,30. Again, I find this unacceptable.
Seven self-citations have been removed, because the original Table 1 has been moved to Supplements.
- Materials and Methods
2.1. Chemicals and Standards
" All chemicals were purchased from Merck (Darmstadt, Germany)." This is not relevant as an subchapter
The sentence has been moved to another subchapter.
2.2. Isolation, Molecular Identification, and Cultivation of Four Strains of Thermophilic Bacteria from Hot Spring
Line 105: how were the samples collected and transported/stored?
Line 107: How was the isolation of the strains performed?
The following part was inserted in the manuscript. “Four strains of thermophilic bacteria were isolated from thermal springs Štěpánka, Sadový, Mlýnský, and Vřídlo in Carlsbad, Czech Republic, and are listed in Table 2S, see Supplements and a previously published paper [3]. Water samples were taken from thermal springs using 50 mL sterile Falcon tubes in three replicates for each sample from the same spring of the same location and were transported to the laboratory at a temperature of at least 50 °C provided by thermal box. The collected water samples were immediately used for enrichment cultivations as follows: Volume of 15 mL of spring water was supplemented with 1.5 mL of diluted YPD (dextrose 20 g/L, peptone 20 g/L, yeast extract 10 g/L, pH 7.0) and cultivated on rotary shaker (150 rpm). Cultivations were performed at the temperature corresponding to the original spring source (i.e. 58 °C in Štěpánka, Mlýnský and Vřídlo and 42 °C in Sadový). After 30 days of cultivation, aliquots of the enrichment cultures were streaked on agar medium to obtain separate colonies. Reasoner’s 2A agar (R2A, HiMedia, Brno, Czech Republic) and Thermus 162 agar (HiMedia, Brno, Czech Republic) supplemented by micro-filtered origin spring water were used. Colonies with different morphologies were further cultivated and identified by 16S rRNA as described before [3].”
2.4. Analysis of Hopanoids by LC-MS
Where is the chromatogram? Please add it.
The chromatogram was given in the manuscript; it is the current Fig. 1. The caption of the figure has just been changed to "Liquid chromatography-mass spectrometry of the peracetylated hopanoids using high resolution NLS and PIS."
Round 2
Reviewer 1 Report
Response and changes are adequate. I do not have further comments.
Reviewer 3 Report
The authors have revised accordingly